# Bactericidal Biodegradable Linear Polyamidoamines Obtained with the Use of Endogenous Polyamines

**DOI:** 10.3390/ijms25052576

**Published:** 2024-02-22

**Authors:** Natalia Śmigiel-Gac, Anna Smola-Dmochowska, Katarzyna Jelonek, Monika Musiał-Kulik, Renata Barczyńska-Felusiak, Piotr Rychter, Kamila Lewicka, Piotr Dobrzyński

**Affiliations:** 1Centre of Polymer and Carbon Materials, Polish Academy of Sciences, 41-819 Zabrze, Poland; asmola@cmpw-pan.pl (A.S.-D.); kjelonek@cmpw-pan.pl (K.J.); mmusial@cmpw-pan.pl (M.M.-K.); 2Faculty of Science and Technology, Jan Długosz University in Czestochowa, 13/15 Armii Krajowej Av., 42-200 Czestochowa, Poland; r.barczynska-felusiak@ujd.edu.pl (R.B.-F.); p.rychter@ujd.edu.pl (P.R.); k.lewicka@ujd.edu.pl (K.L.)

**Keywords:** biodegradable polymers, antibacterial polymers, polyamides, amines, biomaterials

## Abstract

The work presents the synthesis of a series of linear polyamidoamines by polycondensation of sebacoyl dichloride with endogenous polyamines: putrescine, spermidine, spermine, and norspermidine—a biogenic polyamine not found in the human body. During the synthesis carried out via interfacial reaction, hydrophilic, semi-crystalline polymers with an average viscosity molecular weight of approximately 20,000 g/mol and a melting point of approx. 130 °C were obtained. The structure and composition of the synthesized polymers were confirmed based on NMR and FTIR studies. The cytotoxicity tests performed on human fibroblasts and keratinocytes showed that the polymers obtained with spermine and norspermidine were strongly cytotoxic, but only in high concentrations. All the other examined polymers did not show cytotoxicity even at concentrations of 2000 µg/mL. Simultaneously, the antibacterial activity of the obtained polyamides was confirmed. These polymers are particularly active against *E. Coli*, and virtually all the polymers obtained demonstrated a strong inhibitory effect on the growth of cells of this strain. Antimicrobial activity of the tested polymer was found against strains like *Staphylococcus aureus*, *Staphylococcus epidermidis*, and *Pseudomonas aeruginosa*. The broadest spectrum of bactericidal action was demonstrated by polyamidoamines obtained from spermine, which contains two amino groups in the repeating unit of the chain. The obtained polymers can be used as a material for forming drug carriers and other biologically active compounds in the form of micro- and nanoparticles, especially as a component of bactericidal creams and ointments used in dermatology or cosmetology.

## 1. Introduction

The threat of contamination with pathogenic microorganisms is a serious problem not only in the medical and healthcare industry but also in other branches of social activity related to the production, transport, and storage of food. Unfortunately, this problem is growing year by year, mainly due to the excessive widespread use of biocides and the related increasing presence of bacteria resistant to currently used antibiotics and antiseptics [1,2]. The World Health Organization (WHO) has declared that antimicrobial resistance (AMR) is one of the top 10 global public health threats facing humanity [3,4]. Post-implantation bacterial infections are considered a huge problem in modern surgery. Of the total number of clinical complications related to the implantation procedure, infections related to microbiological contamination of the implanted biomaterial are assigned the main role [5,6]. The threats associated with this phenomenon, intensified by the emergence of AMR and the limited ability of antibiotics to eradicate biofilms, force us to undertake comprehensive efforts regarding the use of new alternative therapies, antibacterial biomaterials, and biomaterial-assisted delivery of non-antibiotic therapeutics, such as bacteriophages, antimicrobial peptides, and antimicrobial enzymes [7,8]. Synthetic polyamines containing amide groups with a higher molecular weight than natural ones are particularly interesting due to the wider possibilities of biomedical applications, including substances with strong anti-cancer properties [9]. These compounds are mainly obtained by Michael-type polyaddition with bisacrylamides [10]. Polyamidoamines are particularly interesting in this respect due to their biodegradability and useful physicochemical properties. They are extensively used as carriers in drug and gene delivery. These polymers are synthesized in the form of a linear polymer or as a polyamidoamine (PAMAM) dendrimer [11], which is used as a carrier of many drugs in anti-cancer therapies and targeted drug therapies and as a carrier of contrast and marking substances used in computed tomography (CT) and magnetic resonance imaging (MRI) techniques.

Unfortunately, these highly cationic polymers exhibit quite strong toxicity, limiting their applications.

Significantly less toxic linear polyamidoamines (PAAs) are the aza-Michael polyaddition products of primary monoamines or bis-sec-amines with bisacrylamides [12,13]. These polymers are particularly useful as anti-metastatic drugs [14] and as intercellular nonviral carriers of DNA [15]. The PAAs obtained so far by polyaddition of amines and bisacrylamides are mostly polymers that are soluble in water or which form hydrogels. Due to the limited possibility of selecting monomers, their chains are always built of repeating units containing bis-amide derivatives and tertiary amines. Due to such a structure, these polymers are most often difficult to degrade in the conditions of the human body, and some of the degradation products are not biocompatible [12]. For this reason, various attempts have been made to overcome this problem. Monsalve et al. (2010) synthesized several polyamidoamines with very low molecular weight in the reaction of ethyl acrylate with a diaminopropane derivative using lipase as a catalyst [16]. In another study, the use of hydroxyproline was found to be an interesting method for the synthesis of PAAs [17]. However, improvement of the biocompatibility and biodegradation efficiency of this type of polymer may be mostly obtained by using endogenous polyamines (spermine and spermidine) and derivatives of organic acids originating from the human body for their synthesis. This is the main goal of the presented study. The new class of polyamidoamines containing repeating units in the chain composed of derivatives of endogenous amines and sebacic acid (a substance normally occurring in the form of esters in lipids found in sebum and other skin secretions) is presented in this study. Sebacic acid is also an active component of many biochemical pathways in humans [18,19].

## 2. Results

### 2.1. Initial Attempts at the Synthesis of Polyamidoamines

According to the assumptions, a series of linear PAAs with an average molecular weight of not less than 5000 g/mol were obtained using the polycondensation reaction of endogenous polyamines with selected dicarboxylic acid derivatives. In the initial phase of the research, based on the experiments conducted, the optimal composition of the reaction mixture and the method of carrying out the planned reaction were selected. To obtain a linear polymer, prior protection of the secondary amine groups in the polyamines used as monomers is required. Due to the relatively high susceptibility to thermal decomposition of the polyamines selected for the reaction, especially their derivatives with protected secondary amine groups, as well as due to the difficulties in selecting a universal solvent for the monomers and the products obtained, it was necessary to carry out the polycondensation reaction according to the interfacial polymerization technique [20]. Moreover, based on preliminary tests, it was observed that the dicarboxylic acid derivative used in polymerization should contain a fairly long aliphatic chain containing at least five or six methylene groups. As a consequence, such PAAs are soluble in most traditional organic solvents and may be melted at a temperature much lower than their decomposition temperature.

### 2.2. Preparation of Monomers—Synthesis of Polyamines with Protected Secondary Amine Groups

To obtain diamines with blocked secondary amine groups, a three-stage modification process of selected polyamines—norspermidine (N1-(3-Aminopropyl)propane-1,3-diamine), spermidine (N1-(3-Aminopropyl)butane -1,4-diamine), and spermine (N1, N4-Bis(3-aminopropyl)butane -1,4-diamine)—was carried out. In the first step, the primary amino groups in these compounds were blocked by reacting them with benzaldehyde to form imine bonds (R_3_R_2_C=R_1_), which are reversible and easily hydrolyzed in an acidic condition. As a result of the condensation of selected polyamines with benzaldehyde, the disappearance of the original signal coming from the proton -H_2_NCH_2_CH_2_- and the formation of a new signal coming from the proton of the newly formed imine Bz-C=NCH_2_ on the ^1^H spectra was observed. This is illustrated by the following: for norspermidine, Figure 1(A1) (a signal a at 1.51 ppm) and, after the reaction, Figure 1(B2) (a signal at 3.1–3.7 ppm); for spermine, Figure 1(B1) (a signal at 2.76 ppm) and Figure 1(B2) (a signal at 2.5–3.9 ppm); and, for spermidine, Appendix A (a signal at 2.76 ppm) and Appendix A (a signal at 3.0–3.6 ppm). Moreover, all these spectra show signals in the range of 7–8 ppm coming from the attached benzyl-protecting group. A proton signal from the unreacted Bz-OH aldehyde, which should occur around δ = 10 ppm, was not observed. At the same time, the new signal of protected norspermidine and spermidine around δ = 4 ppm appeared, which was attributed to the protons of the group resulting from the side reaction of intramolecular cyclization of the amine chain to form a six-membered hexahydropyrimidine derivative (signal H, Figure 1(Ib) and Appendix A).

This cyclization reaction was previously observed by Culf’s team [21] during the process of protecting the amino groups of norspermidine with salicylaldehyde. According to our estimates, based on the relative intensity of NMR signal H—a characteristic resonance at ca. δ = 4 ppm of the methine proton of the cycle (Figure 1(A2) and Appendix A)—about 37% of the obtained compounds contained this cycle group. Moreover, this cyclic product practically did not affect the course of the further stage of the planned synthesis, because, at a later stage of the protection of secondary amino groups, it was easily regenerating the primary and secondary amino groups.

^1^H NMR spectra of the obtained compounds are presented in Figure 2 and the Appendix A. After the process, a characteristic *g* signal was observed coming from the methyl groups of the formed carbamate group (Boc group) at δ = 1.44 ppm for the norspermidine derivative and 1.45 ppm for the spermidine derivative. Moreover, signals coming from the protons of the benzyl group were still visible, indicating the stable protection of the primary amino groups.

There was no signal H attributed to the presence of a proton coming from a hexahydropyrimidine derivative formed during the side reaction of their intermolecular cyclization on the spectra of polyamines with all amino groups protected (Figure 2a and Appendix A).

In the next stage of the synthesis, the obtained compounds had to be subjected to selective deprotection of primary amino groups. Hydrolysis of imine bonds was performed in the presence of dichloroacetic acid. The deprotection efficiency was estimated using the ^1^H NMR spectra. In the example spectrum of norspermidine after deprotection (Figure 2b), a new signal was observed around δ = 10 ppm coming from the proton of the released benzaldehyde HOBz, and a series of signals originating from its benzyl group (7.54–7.89 ppm) were also observed. These signals were shifted compared to the signals of the benzyl ring present in the derivative with protected primary amino groups (Figure 2a, 7.39–8.27 ppm). After the deprotection of these groups, the remaining a, b, and c signals shifted slightly, and the g signal of -CH_3_ protons of the secondary amine-protecting group (Figure 2b, 1.44 ppm) remained unchanged. The process of deprotection of primary amine groups in the remaining polyamines proceeded similarly.

### 2.3. Synthesis and Properties of Linear Polyamidoamines

The previously obtained and purified derivatives of the polyamines with protected secondary amine groups, as well as putrescine, were used for a polycondensation reaction with sebacoyl dichloride. The interfacial reaction was carried out at the phase boundary: water/chloroform at room temperature (Figure 1). The optimal reaction conditions were determined by modifying data reported in the literature [20,22,23] with the help of additional experimental tests. The reaction was carried out for 1 h, regardless of the type of amine, at room temperature. ^1^H NMR and FTIR spectra of the obtained polymers are shown in Figure 3, Figure 4, and Appendix A.

The presented FTIR spectra of polyamides (Appendix A) obtained in the reaction of putrescine with sebacic dichloride and other polyamidoamines with a protected amino group (Figure 4(Ia,IIa) and Appendix A) show characteristic bands typical of polyamides: these are absorption bands at 3298 cm^−1^ related to stretching vibrations for the NH group; amide I bands, with C=O deformation vibrations occurring around 1694 cm^−1^; amide II bands at 1537 cm^−1^ related to N-H deformation vibrations coupled with C-N stretching vibrations; and amide III bands at around 1300 cm^−1^ corresponding to the coupled deformation vibrations of the NH bond, stretching vibrations of the C-N bond, and stretching vibrations of C-C=O. They confirm the formation of amide bonds and polyamide chain structures, as well as ^1^H NMR spectra (signal d) and the presence of the chain sequences originating from both sebacic acid (signals A, F, and E) and the used polyamines (signals a, b, and c). Under the polymerization conditions, the protection of the secondary amino group was generally stable. However, a detailed analysis of the NMR spectra of polyamidoamines obtained with norspermidine showed that, in this case, a certain part of the amino groups undergoes self-deprotection. Analyzing the relative intensity of the signals associated with the protons of methylene groups adjacent to the blocked amino group (Figure 3(A1); signals b and c), their intensity was lower than theoretically expected, and at the same time we observed a slightly higher intensity of the F signal associated with the methyl groups of the sebacic acid derivative. This effect probably is caused by the presence of methylene groups in the vicinity of the deblocked amino group (Figure 3(A2); signals b′ and c′). For this reason, we observed an increase in the intensity of the F signal (Figure 3(A1); signal F + b′ + c′).

Unfortunately, attempts to use the GPC chromatography technique to determine average molecular weights did not allow for obtaining reliable results, due to difficulties in selecting the solvent and measurement conditions. Based on the determined viscosities of polymer solutions in THF, it was only possible to estimate the values of the average viscosity mass (Mv) of the obtained polymers, using previously obtained data. The parameters of the Mark–Houwink–Sakurada equation determined for N-trifluoroacetylated nylon 6 (a polyamide with a structure slightly similar to the object of our research) in THF solution were used in the calculations [24]. However, the calculated M_v_ values in Table 1 should be treated as indicative data; they amount to approximately 20,000 g/mol. Table 1 also includes the glass transition temperatures (second run) and melting of the semi-crystalline phase (first run) of the synthesized polymers determined using DSC measurements (Appendix A). The obtained DSC thermograms of PAAs with protected amino groups were essentially very similar to the corresponding long-chain aliphatic polyamide thermograms [25,26]. The T_g_ of all polyamidoamines with blocked amino groups was relatively low, around −25 °C to −35 °C. For polymers with no blocking groups, like in the case of polyamides obtained with putrescine, this temperature was higher (Table 1).

All samples showed one endothermic peak with a broad shoulder pointing towards the lower temperature. The appearance of double melting peaks for polyamidoamines obtained with Boc-spermidine and Boc_2_-spermine can be attributed to the existence of two crystalline phases, similar to the polyamide obtained in the reaction of octadecanedioic acid with diaminodecane [25].

To examine the degree of hydrophilicity of their surfaces, thin films were produced from the obtained polyamidoamines with protected secondary amines. All materials obtained were hydrophobic, showing a contact angle of approximately 80° (Appendix A, Table 1).

The last stage of the synthesis was the deprotection of secondary amino groups in the final obtained polyamides by hydrolysis of carbamide bonds carried out in a chloroform solution in the presence of hydrochloric acid. The efficiency of the deblocking reaction was estimated using the obtained ^1^H NMR spectra. The structure of the obtained polyamidoamines was also confirmed using FTIR measurements. Figure 3 shows the changes that occurred in the ^1^H NMR spectra as a result of this process in a sample of polyamidoamines obtained in the reaction of Boc-norspermidine and Boc_2_-spermine. The practical disappearance of the g signals of methyl protons (1.47 ppm) presented in the protecting amine t-butyl group (Figure 3(A1,B1)) after removing the protection of secondary amine groups in polymers was the crucial aspect of this reaction. The spectra of polyamidoamine samples taken before their purification show g′ signals (about 1.1 ppm) coming from the methyl groups of pivalaldehyde released in the reaction of deprotection of the secondary amino groups. Before and after deprotection, the presence of a *d* signal of approximately 5.98 ppm, originating from the proton of the amide group -NH-CO-, was observed. Weak *e* signals related to amino groups, previously assigned to the protons of the secondary amino group in the protected polyamines, also appeared (Figure 1(A1,B1) and Appendix A).

Testing polymers after the deprotection of amino groups in the obtained polymers, it was difficult to find an appropriate solvent for NMR measurements. The synthesized polyamidoamines (PAAs) contain aliphatic hydrophobic segments and hydrophilic segments in the chain. When these polymers are dissolved in a non-polar solvent (chloroform), they probably form micellar structures, which causes a strong suppression of ^1^H NMR signals related to the protons of the hydrophilic segments (Figure 3(B2)). In turn, in polar solvents like (DMSO + H_2_O), the intensity of proton signals connected with hydrophilic segments increases in the NMR spectrum of the same polymer, but some of the proton signals of the hydrophobic segment weaken. A similar phenomenon was previously observed and described for amphiphilic polymers with the ability to self-assemble micellar structures [27,28]. This phenomenon is particularly visible when the hydrophilic segment is the longest in the case of polymers containing repeating units of spermine derivatives containing two amino groups in the chain. Moreover, in the ^1^H NMR spectra, very weak *g* signals of approximately 1.4 ppm assigned to protons of t-butyl groups can still be observed, which indicates that some of these groups have not been deprotected. However, in all polymers, the number of blocked groups does not exceed 10% of the total secondary amino groups.

The effectiveness of unblocking amino groups in the obtained polyamidoamines was confirmed also by FT-IR measurements and analyzing changes in the absorption spectra of the bands obtained before and after deprotection. Figure 4I pictures the spectra of polymers obtained with spermine before and after deprotection of the amino groups. They confirm the effectiveness of unblocking amino groups. After deprotection, a characteristic strong band was observed at the absorption of 3440 cm^−1^ caused by N-H stretching vibrations, which confirms the presence of secondary amino groups in the obtained PAAs. In the case of other polyamide esters (Figure 4II), these bands are not so distinct, because in this range there are bands related to the stretching vibrations of the N-H amide bonds at 3320 cm^−1^.

The ratio of amide to amino groups in polymers containing repeating units of the chain with spermidine or norspermidine derivatives is 2:1 (for comparison, in a polymer with units derived from spermine, it is 2:2); therefore, the bands associated with the presence of amino groups are less visible (Figure 4(IIb)). The successful unblocking of amino groups is also evidenced by the following:-The presence of a signal associated with stretching vibrations in the range of the 2779 cm^−1^ characteristic band in aliphatic amines of secondary and tertiary origin;-The presence of the first amide band (from the second-row amide groups), with C=O deforming vibrations occurring around 1670 cm^−1^;-Signal decay 1663 cm^−1^ vibrations from carbamide groups after unblocking the amino groups;-Formation of a band at 1543 cm^−1^ originating from amines caused by N-H deforming vibrations. This is a low-intensity band that overlaps the band originating from the II amide band in the case of polyamines composed of spermine derivatives (in which there are two amine groups in the unit repeating) after unblocking. This increase in intensity is most visible;-The presence of a signal at 1188 cm^−1^, with C-N stretching vibrations characteristic of secondary aliphatic amines. The observed band at 1650 cm^−1^ is probably due to associated amines.

Most striking was the change in the wettability of the surface of the obtained PAAs after deprotecting the secondary amine groups present in the chain (Table 1). This effect was visible in contact-angle measurements (Appendix A). The obtained polyamidoamines with blocked amino groups had a contact angle of approximately 78° to 80°, so their surface was highly hydrophobic. However, after deprotection, the contact angle for polymers obtained with norspermidine or spermidine was approximately 20°; so, the surface of the samples became strongly hydrophilic. In the case of the PAA sample obtained with spermine (Appendix A), due to the exceptionally strong wettability of the sample surface, accurate measurement of the angle after deprotection was impossible. A water droplet, after falling on the surface, immediately spread on it (initial angle wetting less than 10°). In this case, the presence of two amino groups in the repeating unit of this polymer was responsible for such a strong effect.

After removing the amino-protecting groups, changes in the thermal properties of these polymers were also noted (Table 1). There was an increase in the glass transition temperature (Tg), the largest (from −35 °C to 2 °C) for the polymer sample obtained with spermine containing two amino groups in the repeating unit. This observation confirms that t-butyl groups were responsible for the low glass transition temperature (T_g_) of the PAA with blocked amino groups. DSC thermograms of the first run for all tested polyamidoamines after removing the blocking groups showed an increase in the melting temperature of the crystalline phase to approximately 130 °C and the presence of only one strong and quite narrow endotherm of its melting (Appendix A). A significant increase in the heat of melting was also noted, indicating an increase in the share of the crystalline phase for all samples except the polymer obtained with spermine. The thermograms of these PAAs were very similar to the thermograms of typical aliphatic polyamides [25,29], including the polyamide obtained with putrescine (Appendix A). It should be noted, however, that the melting point of the crystalline phase (Tm) for polyamidoamines was approximately 100 °C lower than the melting point of polyamides.

### 2.4. Assessment of Cytotoxicity of the Obtained Polyamidoamines towards Skin Cells

The next stage of the described research was to use the obtained PAAs in the formation of carriers of biologically active substances used as an ingredient of dermatological or cosmetic antibacterial creams and ointments. For this reason, human skin cell lines were selected for cytotoxicity assessment: fibroblasts (WI-38) and keratinocytes (HaCaTs). Figure 5 shows the effect of the presence of extracts from the obtained polymers on the proliferation of human fibroblasts.

Cells cultured under standard conditions in the presence of the polyamide extract obtained with putrescine (PA) in the concentration range of 0.16–2000 µg/mL and with spermidine (PAA2) in the concentration range of 0.16–1000 µg/mL demonstrated proliferation comparable to the control group and therefore did not show any toxicity. Interestingly, the effect of stimulating cell growth, practically in the entire range of the tested concentrations, was demonstrated by a polymer containing a spermidine derivative (PAA2). However, in the case of the polymer obtained using spermine (PAA3), a reduction in cell viability was noted below 60% of the viability of cells cultured in the control group at concentrations of 62.5–2000 µg/mL and for the polymer obtained using norspermidine (PAA1) in a concentration range of 500–2000 µg/mL.

Figure 6 shows the results of tests on changes in the viability of keratinocytes cultured in the presence of extracts of the same polymers.

For the polyamide synthesized using putrescine (PA), regardless of the concentration, no cytotoxic effect was proved. In several cases, a reduction in cell proliferation was observed after incubation in a medium containing high concentrations of polymer extracts obtained with norspermidine (PAA1) (500–2000 µg/mL), spermidine (PAA2) (1000–2000 µg/mL), and spermine (PAA3) (250–2000 µg/mL). The effect stimulating the growth of keratinocytes occurred only in the case of polyamide samples obtained with putrescine, and only in high concentrations.

To sum up, it was observed that polymers obtained with spermine (PAA3) and norspermidine (PAA1) showed strong cytotoxicity (especially the ones obtained with the norspermidine derivative—the only non-endogenous polyamine used), but in high concentrations. It is worth highlighting that, quite unexpectedly, polyamidoamines showed greater toxicity towards fibroblasts than keratinocytes.

### 2.5. Preliminary Assessment of the Antibacterial and Antifungal Activities of the Obtained Polyamidoamines

According to the assumptions of the conducted research, the presented group of synthesized polymers containing amine and amide groups in the chain, obtained with the use of endogenous amines, should demonstrate not only good biocompatibility but also strong bactericidal activity against a wide spectrum of strains. It is known from previous research that polyamines such as spermine or spermidine, and especially synthetic linear polyamines with high molecular weights, show antibacterial activity against, for example, *S. aureus* clones resistant to antibiotics [30].

Analyses of the antibacterial and antifungal activity of the examined PAAs on selected strains were carried out at polymer concentrations of 0.1 mg/mL for 24 and 48 h. Figure 7 shows the growth of the *Pseudomonas aeruginosa* strain. The presence of all polymers resulted in a decrease in the numbers of this bacterium. The greatest growth inhibition was observed in the sample containing the polyamidoamine obtained using spermine (PAA3) (the chain unit contained two amino groups) after 48 h (strain concentration decreased from 7.35 to 4.68 log_10_ cfu/mL), and, unexpectedly, in the sample of the polyamide (PA) obtained using putrescine, the strain concentration decreased to 4.62 log_10_ cfu/mL.

Results for the antibacterial activity of the obtained polyamides against *Staphylococcus aureus* are illustrated in Figure 8.

The strongest inhibition of the growth of this strain was observed in the case of 24 h contact with the polyamide prepared with putrescine (PA). The number of cells was only 3.8 log_10_ cfu/mL. After 48 h, this value increased significantly to 6.87 log_10_ cfu/mL but did not exceed the level of the control sample (7.8 log_10_ cfu/mL). Polyamidoamines synthesized with spermine (PAA3) also had a strong antibacterial effect, regardless of the incubation time (decrease from 7.8 to 4.63 log_10_ cfu/mL). After 48 h of incubation, only the polyamidoamine obtained with norspermidine (PAA1) showed almost no antibacterial activity. Figure 9 demonstrates the results of the antibacterial activity of the synthesized polymers against the Staphylococcus epidermidis strain.

In this case, the polyamide obtained with putrescine (PA) was inactive. All other polyamidoamines showed a strong effect, with a very active polymer containing a spermine derivative (PAA3) presenting a particular effect; after 48 h, a decrease in the concentration of cells from 9.0 log_10_ to 3.3 log_10_ cfu/mL was recorded.

Figure 10 shows the antibacterial activity of the tested polymers against the *E. coli* strain. Virtually all samples showed high activity with respect to inhibiting cell growth, regardless of the exposure time. The strongest growth inhibition effect was determined for the PAA3 polymer sample obtained with spermine after 24 h. The number of *E. coli* cells decreased to 4.86 log_10_ cfu/mL, and after 48 h there was a slight further decrease in their number to 4.68 log_10_ cfu/mL.

The antifungal activity of the obtained polyamidoamines against two selected strains largely responsible for the most common clinical cases of candidiasis [31] and aspergillosis [32] are demonstrated in Figure 11 and Figure 12. Figure 11 illustrates the decrease in the number of *Candida albicans* cells after 24 h and 48 h of contact with the tested polymers.

The growth inhibition of this strain was the strongest in the culture carried out in contact with a sample of the polyamidoamine obtained with norspermidine (PAA1). After 48 h, a decrease to 4.52 log_10_ cfu/mL was recorded. In the remaining samples, after 24 h, a temporary inhibition of the growth of these fungi was noted, while, after another 24 h, the cells multiplied, even exceeding the number of cells in the control sample. The next graph (Figure 12) shows the results of activity tests against the *Aspergillus brasiliensis* strain. In this case, the greatest decrease in the number of cells was observed after contact with a PAA3 sample of the polymer obtained with spermine after 24 h (2.59 log_10_ cfu/mL), and, after another 24 h, a slight increase in their amount was noted (3.56 log_10_ cfu/mL). For the remaining samples, the obtained values exceeded the values of the control sample.

To sum up, the antibacterial effect of the obtained polyamides was in many cases strong. This was particularly visible in the case of *E. coli*, where practically all polymers strongly inhibited the growth of cells of this strain. The broadest spectrum of activity was demonstrated by the polyamidoamine PAA3 obtained with spermine and, surprisingly, the polyamide PA obtained with putrescine against *S. aureus*. Polymers obtained with spermine and putrescine also showed fungicidal properties. However, this activity was much lower than the antibacterial effect. The special activity of the polyamidoamine obtained from spermine can be explained by the greater number of amino groups in this polymer. The activity of the polyamide obtained with putrescine, i.e., without amino groups, only containing amide groups, is difficult to explain at this stage of research. However, it proves that the amide groups in the polymer are not neutral and enhance the antibacterial activity of amines, similarly to the previously described composite of polyamide11 and a guanidine derivative [33], or fibers formed from polyamides grafted by ammonium derivatives [34].

## 3. Discussion

Linear polyamidoamines can be obtained by polycondensation of endogenous polyamines with sebacoyl dichloride via an interfacial reaction. Some difficulties are caused by the preparation of monomers and the protection of secondary amine groups of polyamines, which is necessary before polymerization. The polymerization itself using the method described in this work is relatively easy and quick, and the protection of the secondary amine groups of the monomers was stable throughout the polymerization period. The performed synthesis procedure allows for obtaining linear, soluble polymers with a melting point lower than the decomposition temperature. There are no noticeable amounts of by-products in polycondensation products. During polymerization, there is practically no control over the process, which makes it seem impossible to obtain high-molecular polyamidoamines. For this reason, the obtained polymers will not be suitable for forming implants or scaffolds for tissue culture, but they can be used as an interesting material in the formation of biodegradable antibacterial drug carriers in the form of micro- and nanoparticles, or in the production of temporary antibacterial coatings.

We confirmed that the obtained polyamidoamines synthesized using endogenous amines demonstrate relatively low toxicity. Comparing the cytotoxicity of the polymers obtained from norspermidine and spermidine, which are very similar in terms of chain structure and properties, we noticed that the spermidine-based PAA obtained with the use of endogenous polyamine is much less cytotoxic when compared to the other one. The observed cytotoxic effect on the tested cell lines of human skin cells occurred only at high concentrations of the polymer containing norspermidine derivatives in the repeating chain units and the polymer containing spermine derivatives. In the latter case, the reason is clear, since this polyamidoamine contains two amino groups in the repeating unit of the chain, while the others contain only one. It can be seen that the toxicity of this type of polymer is determined by the presence of amines, as evidenced by the lack of cytotoxicity of the polyamide obtained from putrescine. The toxicity effect also depends on the type of cells. Human keratinocytes were significantly more resistant to contact with the obtained polymers than fibroblasts.

The performed tests showed the bactericidal activity of the synthesized polymers. The polyamidoamines obtained from spermine exhibited particular activity, which can be explained by the larger number of amino groups in the chain of this polymer. Unlike the polyesteramines described in our previous report [35], the polymers obtained from spermine and putrescine also presented fungicidal activity, though this was much weaker than their antibacterial activity. The quite unexpected strong bactericidal action of the polyamide obtained with putrescine, i.e., without amino groups, against *Staphylococcus aureus* is difficult to explain at this research stage, and therefore a much more comprehensive study is required. However, it proves that the amide groups in the polymer are not neutral and enhance the antibacterial activity of the amines, similarly to the composite of polyamide11 and a guanidine derivative [33], or fibers formed from polyamides grafted by ammonium derivatives [34].

In our opinion, further, more detailed studies devoted to biogenic amine-based polymers on the optimization of synthesis conditions, rates of biodegradation, and amphiphilic properties optimization should be performed because these materials are very promising from a cosmetology and dermatology point of view. They can be used, for example, as micro- or nanocarriers of biologically active substances, which are components of antibacterial creams or ointments, as well as dressings for non-healing wounds.

The use of polyesteramides in the form of nanocarriers of selected antibiotics described in this work, due to the synergistic effect, may also be an effective weapon in the fight against drug-resistant bacteria [36,37,38].

## 4. Materials and Methods

### 4.1. Materials

Monomers: Putrescine, Butane-1,4-diamine (Sigma-Aldrich, Saint Louis, MO, USA), Norspmidine, Spemidine *N*^1^-(3-Aminopropyl)butane-1,4-diamine, Spermine, *N,N′*-bis (3-aminopropylo)butano-1,4 diamino, Sebacoyl dichloride.

Auxiliary substances: Benzaldehyde (Merck KGaA, Feltham, UK); Di-*tert*-butyl dicarbonate (Merck KGaA, Darmstadt, German); tetrahydrofuran, anhydrous (Avantor, Gliwice, Poland); Dichloroacetic acid (Sigma-Aldrich); Trifluoroacetic acid (Sigma-Aldrich).

### 4.2. Polyamine Modification Procedures

#### 4.2.1. Procedures for the Protection of Primary Amine Groups in the Polyamines

Selective protection of primary amino groups in norspermidine, spermine, and spermidine was carried out using reactions with benzaldehyde according to a modified method described in the literature [21,39]—converting amines to imines by forming a Schiff base.

In a three-neck flask with a capacity of 25 mL equipped with a Dean-Stark cap, a reflux condenser, and a magnetic stirrer, 0.038 mol, V = 5 mL, of norspermidine was placed and then dissolved in 1 mL of anhydrous toluene. After dissolution, 0.076 mol benzaldehyde was added, V = 8.06 mL. The molar ratio of norspermidine to aldehyde used was 1:2.1. The reactions were heated at 100 °C. The synthesis was carried out until the separation of water in the azeotropic cap ended. The reaction time was approximately 3 h. Selective protection of amino groups occurred under azeotropic conditions. After cooling the reaction mixture, the solvent was evaporated and dried to give a viscous yellow liquid.

In the case of protection of primary amino groups in spermidine and spermine, the reactions were carried out analogously.

Spermidine was dissolved in anhydrous toluene (m = 5 g, *n* = 0.035 mol) and then benzaldehyde was added, V= 7.46 mL, *n* = 0.07 mol.

Spermine (m = 2 g, *n* = 0.00988 mol) was dissolved in anhydrous toluene and then benzaldehyde was added, m = 2.096 g, *n* = 0.01976 mol.

#### 4.2.2. Procedures for the Protection of Secondary Amine Groups in the Polyamines

After protecting the primary amino groups in norspermidine, spermidine, and spermine, it was possible to protect their secondary amino groups to obtain more stable blocking groups than was the case when protecting the terminal primary amines. According to a modified procedure described previously [40], this process was performed by reaction with di-t-butyl carbonate (BOC)_2_O (Figure 2) for the following:(2a)N-[(1E)-phenylmethylene]-N′-(3-{[(1E)-phenylmethylene]amino}propyl)propane-1,3-diamine;(2b)N-[(1E)-phenylmethylene]-N′-(3-{[(1Z)-phenylmethylene]amino}propyl)butane-1,4-diamine;(2c)N,N′-bis(3-{[(1E)-phenylmethylene]amino}propyl)butane-1,4-diamine.

The reaction set consisted of a three-neck reactor with a capacity of 250 mL, a reflux condenser, and a dropping funnel. The reactions were carried out under azeotropic conditions. In the reactor, bis-benzylideneimino norspermidine (m = 10 g, *n* = 0.0338 mol, M = 295 g/mol) was dissolved in 40 mL of anhydrous THF. After dissolving this norspermidine derivative, di-tert-butyl dicarbonate (DDC) (M = 228 g/mol, *n* = 0.0338 V = 7.716 mL) was added, which was dissolved in 5 mL of anhydrous THF. While the reaction proceeded under intensive stirring, a solution of DDC and THF was slowly added dropwise. The reaction solution was heated to a temperature of 40–50 °C. During the reaction, bubbles were visible, and a precipitate was formed, which dissolved over time. The reaction took approximately 5 h. After cooling the reaction mixture, the solvent was distilled off on a rotary evaporator. The reaction produced an oily liquid. A similar procedure was followed when blocking (2b), the spermidine derivative. It was dissolved (m = 3.5 g, *n* = 0.00966 g/mol, M = 362 g/mol) in 14 mL of anhydrous THF. Then, once dissolved in the THF, 2 mL was added dropwise (DDC, m = 4.41 g, *n* = 0.01932). In the case of blocking the amino groups of the spermine derivative (2c), the molar ratio of the DDC blocking agent to polyamine was 2:1.

#### 4.2.3. Selective Deprotection of Primary Amine Groups in the Polyamines

To prepare the derivatives of the polyamines (Figure 3(3a–3c)) with protected primary and secondary amines for the polycondensation reaction, the primary amine blocking groups were removed with dichloroacetic acid (DCA) in an aqueous medium. The reaction was carried out under such conditions as to maintain the protection of secondary amino groups in the compounds. Compound (3a) (m = 5 g, *n* = 0.0381 mol, M = 131.22 g/mol) was dissolved in ethylene acetate (AcOEt), 1.4 mL. Then, a DCA/H_2_O mixture (2.8 mL, 50/50% *v*/*v*) was added and stirred for 1 h at room temperature. All organic extracts were repeatedly washed in turn with half their volumes of 1 M aqueous solution KHSO_4_ and then 1 M aqueous solution NaHCO_3_. After washing, the extract was dried over anhydrous MgSO_4_.

Finally, solvents and volatile post-reaction by-products were evaporated from the final extract using a rotary evaporator. Compounds (3b) and (3c) were treated similarly.

### 4.3. Synthesis of Polyamidoamines

A series of polyamidoamines were obtained using interfacial polymerization using previously modified polyamines containing active terminal primary amine groups and protected secondary groups (Figure 1). The interfacial reaction was carried out using previous descriptions [20,22] using two mutually immiscible solvents: water and the organic-phase chloroform CHCl_3_. A measured amount of polyamine (0.05 mol) was placed in a 250 mL round-bottom reactor equipped with a magnetic stirrer and dissolved in water (100 mL). After the amine was completely dissolved, Na_2_CO_3_ (0.1 mol) was added while stirring was continued. Separately, an organic phase was prepared in a separator, consisting of a solution of sebacoyl chloride (0.05 mol) in chloroform (100 mL). The contents of the separator were quickly poured into a reactor containing an aqueous phase with dissolved amines. The synthesis of polyamidoamines was carried out at room temperature, and the contents of the reactor were stirred at a maximum rotation speed. During the reaction, the formation of a layer between the organic and aqueous phases was visible. The process took about 1 h. During the reaction, the salt precipitated in the form of a white precipitate. The mixture was purified by separating the aqueous and solid phases (salt) from the organic phase. The synthesized polymer dissolved in the organic phase was then precipitated in cold diethyl ether. The resulting product was washed with water and dried in a vacuum oven.

### 4.4. Procedures for Deprotecting Amino Groups in Polyamidoamines

Polymers containing carboamine-protected amine groups (2.5 g) were dissolved in dichloromethane (20 mL) and placed in a glass reactor equipped with a stirrer. After complete dissolution, 40 mL of the mixture obtained from HCl and H_2_O was added in a volume ratio of 50:50, and the mixture was stirred. The reaction was carried out at room temperature for 2 h. The solvents and volatile components of the reaction mixture were evaporated on a rotary evaporator. The obtained product was precipitated in cold diethyl ether or hexane and dried to a solid mass.

### 4.5. Measurements

#### 4.5.1. Nuclear Magnetic Resonance (NMR) Spectroscopy

The composition of the polymers was determined with NMR measurements. The 1H NMR spectra of the copolymers were recorded at 600 MHz with the Advance II Bruker Ultrashield Plus Spectrometer (Billerica, MA, USA) and with the use of a 5 mm sample tube. Deuterated DMSO d6 or chloroform was the solvent, and, as the internal standard, tetramethylsilane was used. All ^1^H NMR spectra were obtained with 32 scans, a 2.65 s acquisition time, and an 11 ms pulse at 26 °C. The assignment of signals in the obtained spectra was based on the assignments described earlier [41,42,43].

#### 4.5.2. Thermal Properties

By differential scanning calorimetry (DSC) (using a DuPont 1090B apparatus calibrated with gallium and indium), thermal properties, such as the glass transition temperatures and the heats of melting and crystallization of the obtained copolymers, were examined. The glass transition temperature was determined with a heating and cooling rate of 20 °C/min in the range between −100 and 220 °C, according to the ASTM E 1356-08 standard [44].

#### 4.5.3. Fourier Transform Infrared (FTIR) Spectroscopy

The spectra were recorded in KBr discs in the range of 4000–400 cm^−1^ at 64 scans of samples using a JASCO FT/IR-6700 spectrophotometer (Easton, MD, USA) with a resolution of 2 cm^−1^.

#### 4.5.4. Wettability

Wettability tests were performed on polymeric films prepared by dissolving 0.5 g of each polymer in 10 mL dichloromethane (DCM), pouring them into a 9 cm diameter glass Petri dish, and leaving the solvent to evaporate for 24 h. Tests were performed via the drop shape analysis system (DSA 25, Kruss, Germany) with the use of ultra-high-quality water (UHQ water produced in a UHQ PS apparatus (Elga)) using the sessile drop method. SFE was calculated according to the Owens–Wellt equation using water and diiodomethane (Sigma Aldrich, Germany) as polar and non-polar liquids, respectively. In each case, 10 drops (0.5 µL in volume) were seeded on the surface of the samples and the contact angle was measured automatically.

#### 4.5.5. Determination of Intrinsic Viscosity and Estimated Viscosity Average Molecular Mass

The intrinsic viscosity (η) of the product in THF was determined at 25 °C using an automatic Ubbelohde viscometer to measure the viscosity of polymer solutions in THF. The viscosity average molecular mass (M) was estimated with the Mark–Houwink–Sakurada equation:[η] = KM^a^
where the calculations used previously determined the parameters for nylon 6, K = 1.66 × 10^−2 mL^/g, and a = 0.7 at room temperature in THF [24].

#### 4.5.6. Assessment of Cytotoxicity of the Obtained Polyamidoamines

Cytotoxicity testing was performed following the ISO 10993-5 standard [45]. Human WI-38 fibroblasts (CCL-75), obtained from the ATCC, were cultured in DMEM supplemented with 10% bovine serum (FBS), 100 U/mL penicillin, and 100 μg/mL streptomycin. Human keratinocytes (HaCaTs) were purchased from the Cell Line Service (CLS) and cultured in DMEM supplemented with 10% bovine serum (FBS), 100 U/mL penicillin and 100 μg/mL streptomycin, and 2 mM L-glutamine. A quantity of 10 mM HEPES (pH 7.3) was also added to the experimental cultures. The cells were incubated at 37 °C, 5% CO_2_. Before cell culture, the materials were sterilized with a UV lamp. Each sample was placed in a vial and DMEM was added to obtain a concentration of 1000 μg/mL. The samples were incubated at 37 °C for 24 h. After this time, dilution of the extract was obtained in the concentration range of 0.78–1000 μg/mL. To test for cytotoxicity, 100 μL of the cell suspension, containing 4 × 10^3^ cells, was transferred to the wells of 96-well plates and cultured in standard medium for 24 h to ensure cell adhesion. After 24 h, the medium was replaced with a medium containing the extract of the tested material. Cells were incubated with the tested extracts for 72 h. Untreated cells were used as a negative control (K−), and cells treated with 5% DMSO were used as a positive control (K+). Cell viability was assessed using the Cell Counting Kit-8. Absorbance was read at 450 nm (reference: 650 nm). Statistical analysis was performed using the Statistica 10.0 program, using one-way ANOVA. Results at the significance level of *p* < 0.05 were considered statistically significant.

#### 4.5.7. Assessment of Antibacterial and Antifungal Properties

Assessment of the antibacterial and antifungal activity of the tested samples and inhibitory concentrations were estimated by the microtiter broth dilution method, according to the recommendations of the Clinical and Laboratory Standards Institute (1055 Westlakes Drive, Suite 300 Berwyn, PA 19312, USA) [46]. Samples of each polymer were prepared at concentrations of 20, 10, 1, and 0.1 mg/mL by preparing an aqueous solution (in the case of water-soluble samples) or an aqueous suspension and then rapidly tested. The most interesting results, obtained at the lowest polymer concentration tested, are presented in the work.

Tubes without test compounds were used as positive growth controls. A diluted bacterial suspension was added to each tube to obtain a final concentration of 5 × 10^5^/5 × 10^6^ colony-forming units (cfu)/mL, as confirmed by the number of viable cells (determined by turbidimetry). A bacterial inoculum was used as a negative growth control. The plates were incubated at 37 °C for 24 and 48 h. The contents of the tubes showing no visible growth were plated on selective media and, after overnight incubation at 37 °C, the number of colonies was counted. At least three independent determinations were made for each strain and the modal value was taken. The following strains were selected for testing:Gram-positive: *Staphylococcus aureus* NCTC 10788/ATCC 6538;Staphylococcus epidermidis NCTC 13360/ATCC 1222;Gram-negative: *Escherichia coli* NCTC 12923/ATCC 8739;*Pseudomonas aeruginosa* NCTC 12924/ATCC 9027;Fungi: *Aspergillus brasiliensis* NCPF 2275/ATC C 16404;Yeast: *Can-dida albicans* NCPF 3179/ATCC 10231.

All strains were obtained from Biomaxima S.A., Centrum Mikrobiologii Biocorp, Poland.

## Data Availability

The data presented in this study are available on request from the corresponding author.

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
