# Peer review of "Bactericidal Biodegradable Linear Polyamidoamines Obtained with the Use of Endogenous Polyamines"

_ijms, 2024, doi:10.3390/ijms25052576_

Round 1
Reviewer 1 Report
Comments and Suggestions for Authors
This manuscript describes the synthesis of various linear polyamidoamines and their evaluation regarding their toxicity against eukaryotic cells and bacteria, which may be useful for the screening of antibacterial compounds, but there are still some limitations:
1. The authors should revise the NMR characterization of all compounds as the assignment of the NMR peaks is not correct, e.g. the signal a of norspermidine and spermidine is not at 1.38 and 2.62 ppm, respectively, but at 2.51 and 2.76 ppm, respectively, while in the protected amines the signal a should be ~3.7 ppm and not at 2-2.5 ppm, etc.
2. From NMR spectra of Fig.1 and 2, it is obvious that the products are not pure, which it is expected as the authors did not perform any purification process after the reaction. The products received after solvent evaporation. Probably they received expect of the desire products some by-products as mono-substituted amines. These by-products will affect the subsequently synthesis of polyamines as dendritic polyamines could be synthesized. Thus, the protected amines should be purified and characterized again by NMR. Also, the degree of protection should be calculated by the integration of the peaks.
3. Lines 134-138. The authors claimed that the cyclic compound is 5-7% of the total product. How did they calculate it? From the spectrum (I)b in Figure 1
4. Figure 1 caption: “II) …. b) after deprotection;” The correct is after protection.
5. The structure of BOC group is not correct in the Figure 2.
6. The peaks of spectra presented in Figure 3 are too small, so it is impossible to be receive any information for the structure of polymers.
7. Lines 695-696. It is not clear the procedure of extract preparation. Also, I cannot understand the reason that the authors measured the toxicity of these extracts and not the toxicity of the polymers. It is known that an antibacterial agent should exhibit simultaneously antibacterial properties and low toxicity in order to be useful for further applications. Thus, the synthesized polymers should be evaluated regarding their toxicity and not their extracts. For this reason, the same concentrations of the aqueous polymeric solutions as those at the antibacterial experiments should be used.
8. The unit of y-axis is not correct in Figures 7-12. Shoul be log10(cfu/mL).
The authors should carefully polish the English writing throughout the manuscript. There are several type- errors both in the manuscript and the supplementary material that should be corrected.
Author Response
Response to reviewer's 1 comments.
Thank you very much for your insightful review.
This manuscript describes the synthesis of various linear polyamidoamines and their evaluation regarding their toxicity against eukaryotic cells and bacteria, which may be useful for the screening of antibacterial compounds, but there are still some limitations:
- The authors should revise the NMR characterization of all compounds as the assignment of the NMR peaks is not correct, e.g. the signal a of norspermidineand spermidine is not at 1.38 and 2.62 ppm, respectively, but at 2.51 and 2.76 ppm, respectively, while in the protected amines the signal a should be ~3.7 ppm and not at 2-2.5 ppm, etc.
Thank you very much for your attention. Indeed, the assignment of the signals in the mentioned spectra appears to have been incorrect. We changed them using previous work describing the 1H NMR spectra of putrescine (Chemical Physics Letters 466 (2008) 219–222.), spermidine and norspermidine (Scientific Reports | (2019) 9:14971 | https://doi.org/10.1038/ s41598-019-50943-1), and spermine (Eur. J. Biochem. 269, 4317–4325 (2002)). We also used a program simulating spectra based on the chemical structure of a compound from the ACD Lab package. We noticed some inconsistency in the assignments of the signals of these polyamines currently present in the literature (basically differences in the assignment of the NH2-CH2 and NH-CH2 proton signals). After a thorough analysis of the literature and the results of our research, we made changes in line with the reviewer's suggestion. We have introduced appropriate changes in the assignment of signals in all presented spectral figures and an appropriate mention in the description of the NMR measurements (experimental part).
- From the NMR spectra of Fig.1 and 2, it is obvious that the products are not pure, which is expected as the authors did not perform any purification process after the reaction. The products received after solvent evaporation. Probably they received expect of the desire products some by-products as mono-substituted amines. These by-products will affect the subsequently synthesis of polyamines as dendritic polyamines could be synthesized. Thus, the protected amines should be purified and characterized again by NMR. Also, the degree of protection should be calculated by the integration of the peaks.
Yes, that's a very good point. In our research, we tried to purify the product as effectively as possible after each stage of synthesis (by washing and separating solutions). However, the most important thing was to properly purify the final polyamine derivatives used as a monomer (containing active terminal primary amino groups), which was practically difficult due to the need to maintain full protection of the amino groups. The spectra shown in Figures 1 and 2 are spectra of compounds obtained before the purification process (to show the products and by-products formed), hence probably the comments of the reviewer. We have added an explanation in the description of this drawing. We did not use the method of purifying amine mixtures using column chromatography described in the literature, due to the high burden of this method when using larger amounts of solution.
Protected amines obtained by reacting amine groups with benzylaldehyde and Boc2O were purified essentially by washing the organic solution several times with aqueous 1 M solutions of bisulfide potassium water solvate and finally NaHCO3. This is quite an old method, used when purifying amine mixtures, recommended in the literature (Selective Protection of Polyamines: Synthesis of Model Compounds and Spermidine Derivatives J. Chem.Soc. Perkin Trns., 1988, 1905 – 1911., Spermidine Derivatives and A Practical Guide for Buffer-Assisted Isolation and Purification of Primary, Secondary, and Tertiary Amine Derivatives from Their Mixture Organic Process Research & Development 2005, 9, 847-852.) The description of the procedure for purifying monomers was supplemented in the experimental part.
To prevent the presence of incompletely blocked derivatives of the amines used, we tried to obtain amine derivatives with a maximum degree of protection of the amine groups. For this reason, we carried out amine protection reactions for a relatively long period and with a slight stoichiometric excess of the blocking agent. We continuously monitored the progress of the reaction by analyzing NMR spectra.
When it comes to selective deprotection reactions, there were indeed problems with full deprotection of the secondary amine groups of the final polyamidoamines, as we mention in the text of the manuscript. Complete deprotection of these groups was not achieved due to the fear of causing excessive degradation of the polymer.
- Lines 134-138. The authors claimed that the cyclic compound is 5-7% of the total product. How did they calculate it? From the spectrum (I)b in Figure 1
The reviewer rightly pointed out the calculation error. The recalculated content of norspermidine and spermidine derivatives containing a cyclic moiety is much higher and amounts to approximately 35-37%, not 5-7%. Sorry, I don't know where this big mistake came from. The most important thing, however, is that after the stage of deprotection of primary amine groups in polyamines, there are no cyclic products, on NMR spectra that lack this characteristic single proton signal (Figure 2). We have made appropriate changes to the manuscript's text.
The contain of molecules of norspermidine derivatives and spermidine derivatives containing cyclic groups, formed during the process of blocking the amine end groups, was estimated based on the relative intensity of the H - characteristic resonance signal at ca. δ = 4 ppm of methine proton of the cycle (Figure 2b and Figure S1b) concerning the signal intensity of group b CH2 protons of the obtained product - Ccyclic % = [H /(H + ((b – 2H)/4)] x 100
- Figure 1 caption: “II) …. b) after deprotection;” The correct is after protection.
We have corrected this error.
- The structure of BOC group is not correct in the Figure 2.
We have corrected this error, the reviewer's insight must be appreciated - thank you.
- The peaks of spectra presented in Figure 3 are too small, so it is impossible to be receive any information for the structure of polymers.
We re-performed the NMR spectrum of the obtained polyamidoamine using a spermidine derivative; in deuterated chloroform and deuterated DMSO. These spectra are attached to the figure (Figure 3 II b and c). We managed to obtain slightly stronger signals. However, the observations described earlier were repeated, i.e. in chloroform, the signals of the hydrophilic chain sequence were very weak, but the proton signals of the sebacic acid sequence were quite clear. In the spectrum obtained from the deuterated DMSO solution, all signals were practically visible, and their relative intensity was close to expected. We have improved the signal assignment as recommended by the reviewer.
- Lines 695-696. It is not clear the procedure of extract preparation. Also, I cannot understand the reason that the authors measured the toxicity of these extracts and not the toxicity of the polymers. It is known that an antibacterial agent should exhibit simultaneously antibacterial properties and low toxicity in order to be useful for further applications. Thus, the synthesized polymers should be evaluated regarding their toxicity and not their extracts. For this reason, the same concentrations of the aqueous polymeric solutions as those at the antibacterial experiments should be used.
The Authors would like to thank you for the comments and explain the methodology used in the manuscript for in vitro cytotoxicity testing. The cytotoxicity study was conducted strictly according to the ISO 10993-5 standard. The ISO 10993-5 standard recommends performing the test on an extract of the test sample and/or the test sample itself. The extraction method has been widely used for testing the cytotoxicity of biomaterials because it can be applied to a wide variety of raw materials and finished products that may release toxins from exposed surfaces [https://doi.org/10.3892/br.2015.481; doi: 10.3349/ymj.2005.46.4.579; https://doi.org/10.1016/B978-0-12-812258-7.00010-1].
In general, the results of the extracts of medical devices are usually consistent with the results of animal toxicity tests [https://doi.org/10.3892/br.2015.481]. Selection of the extract testing for the materials presented in the manuscript was the most obvious way to analyze the row material, without further processing. Analysis of the cytotoxicity in direct contact with tested material requires its processing because according to the ISO 10993-5 standard, “the preferred test sample of a solid material should have at least one flat surface”. The processing may have an additional impact on the materials, so extraction was selected as a more appropriate way of testing. The extract has been prepared in the culture medium with serum for 24h at °C, which is following the ISO 10993-5 recommendations. The concentration range (2000 – 1.16 mg/mL) of the polymer extract used for the cytotoxicity study involved also concentration (0.125 mg/mL), which was similar to the concentration used for the antibacterial study (0.1 mg/mL).
- The unit of y-axis is not correct in Figures 7-12. Shoul be log10(cfu/mL).
We have corrected this error.
Comments on the Quality of English Language
The authors should carefully polish the English writing throughout the manuscript. There are several type- errors both in the manuscript and the supplementary material that should be corrected.
We corrected the grammatical and lexical correctness of the manuscript text.
Reviewer 2 Report
Comments and Suggestions for Authors
Despite the results and relevance of the work, there are drawbacks in it:
1) In introduction chapter, in the sentence: "Unfortunately, this problem is growing year by year, mainly due to the excessive widespread use of biocides and the related increasing presence of bacteria resistant to currently used antibiotics and antiseptics" the reference is absent.
2) In figures, to highlight dimensions, you use either square or round brackets or do not use them at all. Make it consistent for all figures.
3) In figures from 7 to 12 on growth values and on OY axis’s you use commas. In OY axis you should delete commas and show integer numbers, for growth values use dots instead commas.
4) In general, I think, that the quality of the figures is bad. It seems to me that you are using "Excel" to draw these figures, which is not recommended for a good quality research paper. For this I recommend to use "OriginPro" software, in which you can draw good quality figures. You should also make the figures in the same style.
5) In introduction you say that it’s the problem of antibiotic resistant bacteria is significant. But I didn't notice in the results that you are testing your polyamidoamines against multi-drug resistant or single-drug resistant bacteria. It is better to show how these polymers can inhibit the growth of this type of bacteria.
Comments on the Quality of English LanguageMinor editing of English language required
Author Response
Response to reviewer's 2 comments
Thank you for your review and helpful valuable comments.
Despite the results and relevance of the work, there are drawbacks in it:
- In introduction chapter, in the sentence: "Unfortunately, this problem is growing year by year, mainly due to the excessive widespread use of biocides and the related increasing presence of bacteria resistant to currently used antibiotics and antiseptics" the reference is absent.
To the reviewer's comment, we have included two references [1,2] in the text of the manuscript, which describe the causes of the presence of drug-resistant bacteria.
- In figures, to highlight dimensions, you use either square or round brackets or do not use them at all. Make it consistent for all figures.
We have improved all figures according to the reviewer's comments.
- In figures from 7 to 12 on growth values and on OY axis’s you use commas. In OY axis you should delete commas and show integer numbers, for growth values use dots instead commas.
- In general, I think, that the quality of the figures is bad. It seems to me that you are using "Excel" to draw these figures, which is not recommended for a good quality research paper. For this I recommend to use "OriginPro" software, in which you can draw good quality figures. You should also make the figures in the same style.
We have once again made some of the charts presented in the figures mentioned by the reviewer, using an appropriate professional program. We used dots to replace incorrect commas too.
- In introduction you say that it’s the problem of antibiotic resistant bacteria is significant. But I didn't notice in the results that you are testing your polyamidoamines against multi-drug resistant or single-drug resistant bacteria. It is better to show how these polymers can inhibit the growth of this type of bacteria.
The presented manuscript mainly presents the possibility of obtaining biodegradable polymers with quite strong antibacterial activity. In further research, we plan to check the activity of these materials also against selected drug-resistant bacteria. Generally, however, such an effect can be expected when the polymers described in this work are used as nanocarriers of selected antibiotics, as evidenced by previous studies on similar antibiotic release systems. Only such joint action of both factors can result in a positive result thanks to the previously described synergistic effect. We introduced appropriate commentary into the text of the manuscript in the Discussion section, providing the text with several literature references.
Round 2
Reviewer 1 Report
Comments and Suggestions for Authors
The authors were taken under consideration my suggestion and now the manuscript appears improved. However, there are some issues that have to be considered before publication. Specific points are:
1. Although the authors corrected the NMR peak assignment, they forgot to correct some numbers in lines 164-167. Please correct.
2. Please provide structural characterization (NMR and FTIR) for polyamide obtained using putrescine.
3. Please add in the captions of the figures with NMR spectra what solvent was used.
4. In line 198: spectra of the compounds with protected amino groups are also shown in Figure 2, not only in S2. Please correct.
5. Figure 3. Please show the spectra from at least 10 ppm to 0 ppm in order to confirm the purity of the final polyamides. Also, in spectra I(a) and (b), the peak of a methylene is two strong compared to b and c or c’. As all the peaks correspond to 4 protons should be almost equal. Moreover, in spectrum I(b), the authors said that the peak at 1.72 ppm attributed to A+F methylenes, while the peak at 1.31 attributed to E methylenes, but the first peak is too strong compared to the second one. This is very odd as the proton ratio of A+F to E is 1:2.
The peaks in spectrum II(B) are too weak so it is impossible to be confirmed the polymer structure. Perhaps the solution was very diluted so please repeat this spectrum with more concentrated solution. Perhaps you should use another solvent.
6. Please discuss the NMR peaks related to the newly formed amide groups.
7. FTIR spectra for polyamides obtained used putrescine and nonspermidine are missing.
8. Lines 370-373: How did the authors calculate the ratio of amide to amino groups?
9. Lines 826-828: Please re-write these 2 sentences in order to clarify the preparation method of extracts and how the tested concentrations were calculated. Also check the concentration range because is different than that presented in the results section.
10. In Figures 7,8,9 and 10, the controls of 24h are missing.
11. Line 847: It was written that aqueous polymeric solutions at concentrations of 20, 10, 1, and 0.1 mg/mL were prepared but in the results sections the authors present only results using solutions at 0.1 mg/mL. Please correct. Also please check all the experimental part for the antibacterial/antifungal evaluation. I don’t think that the content in lines 854-855 is correct.
12. Lines 721-722: Please correct the “1mol”. I think you want to write 1M.
Comments on the Quality of English Language
There are some type- errors in the manuscript that should be corrected.
Author Response
|
1. Summary |
|
|
|
Thank you again for your detailed and insightful review of our work. Many of the comments and questions sent are valuable, allowed us to understand certain aspects related to the course of the research and the obtained results, and finally to improve the quality of the presented manuscript.
|
||
|
2. Point-by-point response to Comments and Suggestions for Authors |
||
|
1. Although the authors corrected the NMR peak assignment, they forgot to correct some numbers in lines 164-167. Please correct. The error was corrected by introducing appropriate ppm chemical shift ranges – lines 116 – 119.
2. Please provide structural characterization (NMR and FTIR) for polyamide obtained using putrescine. 1H NMR and FTIR spectra of polyamide obtained with putrescine, following the reviewer's comment, were introduced into the content of the supplement (NMR spectrum Figure S3 and FTIR spectrum Figure S4). Appropriate short commentary and description have been introduced into the text of the manuscript. – lines 180-193.
3. Please add in the captions of the figures with NMR spectra what solvent was used. In the captions of Figures 1, 2, 3 and Figures S1, and S2, information regarding the type of solvent used was introduced, which for these measurements we used deuterated chloroform.
4. In line 198: spectra of the compounds with protected amino groups are also shown in Figure 2, not only in S2. Please correct. Corrected according to the reviewer's comment – line 145
5. Figure 3. Please show the spectra from at least 10 ppm to 0 ppm in order to confirm the purity of the final polyamides. Due to the desire to maintain the readability of the spectra presented in Figure 3 (in many cases difficult to observe), we did not extend the range to the area from 0 ppm to 10 ppm, because in this range not practically readable signals, apart from those related to the used CDCl3 solvent and its impurities. Below, to illustrate this decision, we present the spectra of the polymers in question in the postulated range. (the figures 1,2 and3).
Also, in spectra I(a) and (b), the peak of a methylene is two strong compared to b and c or c’. As all the peaks correspond to 4 protons should be almost equal. Moreover, in spectrum I(b), the authors said that the peak at 1.72 ppm attributed to A+F methylenes, while the peak at 1.31 attributed to E methylenes, but the first peak is too strong compared to the second one. This is very odd as the proton ratio of A+F to E is 1:2.
Yes, of course, we agree with the reviewer. This is a very important note. These signals should be similar in intensity. The problem is related to the signal a (very stretched on Figure 3Ia), the intensity of which appears to be lower than expected. But when we extend the limits of integration from about 3.4. ppm to 3.7 ppm, this condition is practically met. Another problem occurred with the interpretation of the E signal in Figure 3Ia because in this case, this signal is a bit too weak concerning the F signal (the intensity of these signals should be in a 2:1 ratio). The reason is the self-deprotection of some of the amino groups that occurs during the final stage of polymerization. This causes the concentration of groups related to the intensity of signals b and c to decrease. However, signals b' and c' appear, which in turn causes an increase in the intensity of the signal at 1.64 ppm (originally signal F, now we have marked it as F+b'+c' ). Hence, there is the mentioned disturbance of the E: F intensity ratio. We have introduced a short additional comment to the text of the manuscript. Lines 194 – 203, changes in Figure 3 I.
The peaks in spectrum II(B) are too weak so it is impossible to be confirmed the polymer structure. Perhaps the solution was very diluted so please repeat this spectrum with more concentrated solution. Perhaps you should use another solvent. We tried various solvents, but their selection was not very large due to the poor solubility of polyamidoamines in standard solvents used in NMR studies. As we describe in the text of the manuscript, the poor image of the obtained spectra is not only related to problems with the solubility of the tested polymers (polyamidoamines obtained with spermine, although they dissolve completely in chloroform, still give strongly distorted spectra).
Please discuss the NMR peaks related to the newly formed amide groups. The text of the manuscript now includes a description of the origin of the d signal appearing in the spectra, related to the proton of the amide group, and evidence of the formation of polymers containing polyamide chains. – lines 191-193.
6. FTIR spectra for polyamides obtained used putrescine and nonspermidine are missing. These spectra have been introduced in the supplement section with a short commentary in the manuscript text (Figure S4 and S5, lines – 182-190)
7. Lines 370-373: How did the authors calculate the ratio of amide to amino groups? We tried this, but the results are not reliable and quite inconsistent because the intensity of these signals, especially the protons of secondary amino groups, depends very much on the conditions in which the measurement was performed. Therefore, we did not want to present these results in the manuscript.
8. Lines 826-828: Please re-write these 2 sentences in order to clarify the preparation method of extracts and how the tested concentrations were calculated. Also check the concentration range because is different than that presented in the results section. We changed this really unfortunate sentence from: "Preparation of extracts: DMEM (1000 μg/ml) was added to the sample vials and the samples were incubated at 37°C for 24 hours. After this time, solutions were obtained in the concentration range of 0.78 – 1000 μg/ml." in the sentence "Preparation of extracts: Each sample was placed in a vial and DMEM was added to obtain the concentration of 1000 μg/mL. The samples were incubated at 37°C for 24 hours. After this time, dilution of the extract was obtained in the concentration range of 0.78 – 1000 μg/ml." - lines 727 -732.
10. In Figures 7,8,9 and 10, the controls of 24h are missing. We introduced an additional bar showing a control sample of 24h. (Figures 7,8,9 and 10) 11. Line 847: It was written that aqueous polymeric solutions at concentrations of 20, 10, 1, and 0.1 mg/mL were prepared but in the results sections the authors present only results using solutions at 0.1 mg/mL. Please correct. Also please check all the experimental part for the antibacterial/antifungal evaluation. I don’t think that the content in lines 854-855 is correct. Due to the initial lack of knowledge about the actual antibacterial activity of the obtained polymers, we started the research with quite high concentrations, not expecting such a strong activity of the obtained polymers. The aim was to demonstrate the lowest concentration at which a clear effect of the tested polymers is visible. In this work, we only showed the results at the lowest tested concentrations, i.e. 0.1 mg/ml, because these are the most interesting results and allow us to observe differences in the activities of the obtained materials. In the future, we plan to present the results of tests on the activity of selected polyamidoamines and compare them with the activity of other groups of polymers obtained in our laboratory (polyesteramines, chitosan modifications) in a special publication, where the exact MIC concentrations will be determined. We have added an appropriate explanation in the manuscript text (line – 754-755)
12. Lines 721-722: Please correct the “1mol”. I think you want to write 1M. Yes, of course, it was about the concentration of 1M solution, the correction was made. |
||

Reviewer 2 Report
Comments and Suggestions for Authors
Taking into account the clarifications and sent files from MDPI editorial office, the article can be published.
Comments on the Quality of English LanguageMinor editing of English language required.
Author Response
Thank you for your efforts in reviewing our manuscript.
Round 3
Reviewer 1 Report
Comments and Suggestions for Authors
The revised manuscript improved as the authors have taken under consideration almost all my suggestions and now, I think that the manuscript is suitable for publication.
Comments on the Quality of English LanguageI think that the quality of English language is good enough.

figure 1. 1H NMR spectra in CDCl3 of polyamidoamine prepared with norspermidine, before deblocking the amino groups
1